# Use of Human Lung Tissue Models for Screening of Drugs against SARS-CoV-2 Infection

**DOI:** 10.3390/v14112417

**Published:** 2022-10-31

**Authors:** Alexander J. McAuley, Petrus Jansen van Vuren, Muzaffar-Ur-Rehman Mohammed, Sarah Goldie, Shane Riddell, Nathan J. Gödde, Ian K. Styles, Matthew P. Bruce, Simran Chahal, Stephanie Keating, Kim R. Blasdell, Mary Tachedjian, Carmel M. O’Brien, Nagendrakumar Balasubramanian Singanallur, John Noel Viana, Aditya V. Vashi, Carl M. Kirkpatrick, Christopher A. MacRaild, Rohan M. Shah, Elizabeth Vincan, Eugene Athan, Darren J. Creek, Natalie L. Trevaskis, Sankaranarayanan Murugesan, Anupama Kumar, Seshadri S. Vasan

**Affiliations:** 1Australian Centre for Disease Preparedness, Commonwealth Scientific and Industrial Research Organisation, Portarlington Road, Geelong, VIC 3220, Australia; 2Department of Pharmacy, Birla Institute of Technology and Science, Pilani 333031, India; 3Department of Medicinal Chemistry, University of Utah, Salt Lake City, UT 84112, USA; 4The Australian Institute for Bioengineering and Nanotechnology, The University of Queensland, Brisbane, QLD 4075, Australia; 5Disposition and Dynamics, Monash Institute of Pharmaceutical Sciences, Monash University, Parkville, VIC 3052, Australia; 6Victorian Infectious Diseases Reference Laboratory, Melbourne, VIC 3000, Australia; 7Manufacturing, Commonwealth Scientific and Industrial Research Organisation, Research Way, Clayton, VIC 3168, Australia; 8Australian Regenerative Medicine Institute, Monash University, Clayton, VIC 3168, Australia; 9Australian National Centre for the Public Awareness of Science, Australian National University, Canberra, ACT 2601, Australia; 10Responsible Innovation Future Science Platform, Commonwealth Scientific and Industrial Research Organisation, Boggo Road, Dutton Park, QLD 4102, Australia; 11Centre for Medicine Use and Safety, Monash Institute of Pharmaceutical Sciences, Monash University, Parkville, VIC 3052, Australia; 12Department of Chemistry and Biochemistry, Swinburne University of Technology, Hawthorn, VIC 3122, Australia; 13Department of Infectious Diseases, University of Melbourne, Melbourne, VIC 3000, Australia; 14School of Medicine, Deakin University, Pigdons Road, Waurn Ponds, VIC 3216, Australia; 15Land & Water, Commonwealth Scientific and Industrial Research Organisation, Waite Campus, Urrbrae, SA 5064, Australia; 16Department of Health, 189 Royal Street, East Perth, WA 6004, Australia; 17Department of Health Sciences, University of York, York YO10 5DD, UK

**Keywords:** COVID-19, CoviRx.org, therapeutics, drug repurposing, 3D tissue models

## Abstract

The repurposing of licenced drugs for use against COVID-19 is one of the most rapid ways to develop new and alternative therapeutic options to manage the ongoing pandemic. Given circa 7817 licenced compounds available from Compounds Australia that can be screened, this paper demonstrates the utility of commercially available ex vivo/3D airway and alveolar tissue models. These models are a closer representation of in vivo studies than in vitro models, but retain the benefits of rapid in vitro screening for drug efficacy. We demonstrate that several existing drugs appear to show anti-SARS-CoV-2 activity against both SARS-CoV-2 Delta and Omicron Variants of Concern in the airway model. In particular, fluvoxamine, as well as aprepitant, everolimus, and sirolimus, has virus reduction efficacy comparable to the current standard of care (remdesivir, molnupiravir, nirmatrelvir). Whilst these results are encouraging, further testing and efficacy studies are required before clinical use can be considered.

## 1. Introduction

As of March 2022, the ongoing coronavirus disease-19 (COVID-19) pandemic has resulted in nearly 500 million cases, over 6 million attributable deaths, and over 18 million excess deaths worldwide [1,2].

Vaccines against COVID-19 are effective at reducing disease severity, but the reduced efficacy against recent variants of the Severe Acute Respiratory Syndrome Coronavirus-2 (SARS-CoV-2) have raised questions about their ability to protect against future variants without new or improved vaccine formulations [3]. Moreover, global vaccine inequity means that approximately 3 billion people, primarily in low- and lower middle-income countries, have yet to receive a single dose of a COVID-19 vaccine [4]. In addition, hesitancy to be vaccinated or receive boosters, inadequate healthcare system funding, and logistical issues in local vaccine distribution also contribute to not only diminished vaccination rates, but also to the emergence of new variants [5]. Therefore, safe, effective, and affordable therapeutics are urgently needed to treat individuals when they inevitably fall ill with COVID-19.

Vaccines are but one tool in the anti-viral arsenal, with drugs and therapies having an equally significant role to play in the response to the COVID-19 pandemic. Monoclonal antibody (mAb) therapies developed against variants circulating early in the pandemic appear to be less effective against the Omicron variant-of-concern (VOC), with the US Food and Drug Administration (FDA) limiting the use of certain monoclonal antibody therapies for the treatment of COVID-19 infections due to this reduced efficacy [6,7].

Within Australia, there are three drugs approved for the treatment of COVID-19: remdesivir (Veklury; Gilead Sciences), molnupiravir (Lagevrio; Merck Sharp & Dohme), and nirmatrelvir + ritonavir (Paxlovid; Pfizer) [8]. Remdesivir and molnupiravir are nucleoside analogues, causing mutations within the SARS-CoV-2 genome and stalling of genome replication, while nirmatrelvir (PF-07321332) is an inhibitor of the main SARS-CoV-2 protease administered in combination with ritonavir [9,10,11]. Although these drugs show efficacy against severe COVID-19, their high cost, and limited availability restrict access for people in low and lower middle-income countries [12].

The repurposing of licenced drugs-drugs that have been approved for use against a particular indication that show efficacy against another disease–is one of the fastest ways to introduce new therapies into clinical use [13,14]. These drugs have already been through clinical trials to demonstrate safety and have established production pipelines. Accordingly, the pathway to the clinic for the treatment of COVID-19 is significantly shorter than for novel compounds. As there are circa 8000 drugs available in open collections such as ‘Compounds Australia’, repurposing is like looking for a needle in a haystack unless a rigorous down-selection is implemented that involves in silico approaches followed by in vitro assessments. In a related study, MacRaild et al. (2022) have methodically down-selected the top 214 candidates from ~8000 candidates in Compounds Australia collection [15]. This paper reports a follow-on study involving the preliminary in vitro/ex vivo evaluation of the top 10 candidates (from the 214 identified in MacRaild et al. (2022)), and controls (remdesivir and molnupiravir) against the SARS-CoV-2 Delta (B.1.617.2) VOC in a human airway model. In addition, it further reports the secondary assessment of three of the candidates plus two analogues with controls (remdesivir, molnupiravir, and nirmatrelvir) against both Delta and Omicron (BA.1.1) VOC in the same model. This step is essential before further in vivo evaluation and human clinical trials can take place, for the following reasons: (1) considering the opportunity cost, it is not realistic to evaluate hundreds of compounds in animal or human studies, therefore a triage process is required; (2) animal and human ethics committees in well-regulated jurisdictions such as Australia require in vitro efficacy data to reduce animal suffering (refining the experimental design; reducing the number of animals required; replacing some of the evaluation with ex vivo models; known as the ‘3R’ objectives) and risk to human health; (3) such in vitro/ex vivo studies are comparatively inexpensive and quick to perform; (4) it is possible to test a range of drug concentrations and combinations, and have control over variables such as the variant used for infection.

The potential for disagreement and disconnect between antiviral efficacy in cell lines and in human patients, balanced with the need to screen and test drugs for efficacy before human use, means that a middle-ground is required. One possible approach to better bridge the gap between in vitro and in vivo experiments is through the use of ex vivo/3D tissue models. The development and use of 3D tissue models (MicroPhysiological Systems, or MPS) has increased significantly over the last 20 years, with commercial companies providing a range of models corresponding to a variety of tissues [16]. These tissue models have the advantage of incorporating a range of cell types into a more representative structure of human tissues, allowing for the complex interplay between different cell types to occur [17].

The respiratory tract is the main route of infection and transmission for SARS-CoV-2. Three-dimensional models of these tissues are important and useful for screening and testing drugs and therapies against SARS-CoV-2 infection [18,19]. Indeed, similar tissue models have been used previously for the testing of antivirals against influenza virus and rhinovirus, and their suitability for anti-SARS-CoV-2 agent testing has some supporting evidence [20,21,22,23,24].

This study aimed to achieve the following objectives: (1) Validation of the EpiAirway and EpiAlveolar tissue models from MatTek for use in SARS-CoV-2 infection and drug screening experiments; (2) Application of the EpiAirway model for screening of ten licenced drugs against SARS-CoV-2 infection; (3) Use of the EpiAirway model for secondary assessment of the three most promising drugs and two analogues for their efficacy against SARS-CoV-2 Delta and Omicron infection.

## 2. Materials and Methods

### 2.1. Viruses and Cells

The Delta (B.1.617.2; hCoV-19/Australia/VIC18440/2021; EPI_ISL_1913206) and Omicron BA.1.1; hCoV-19/Australia/VIC28585/2021) variants of SARS-CoV-2 were kindly provided by Drs Caly and Druce at the Victorian Infectious Diseases Reference Laboratory. Working stocks were grown in Vero E6 cells (American Type Culture Collection; Manassas, VA, USA), with Dulbecco’s Minimum Essential Medium (DMEM) supplemented with 2% FBS, 2 mM GlutaMAX supplement, 100 U/mL penicillin, and 100 μg/mL streptomycin (all components from ThermoFisher Scientific; Scoresby, VIC, Australia). Diluted inoculum was used to inoculate Vero E6 cells for 1 h at 37 °C/5% CO_2_ before additional media was added to the flask. The flasks were incubated for 48 h before supernatant was centrifuged at 2000× *g* for 10 min to clarify, harvested and stored in 1 mL aliquots at −80 °C.

Identity of virus stocks were confirmed by next-generation sequencing using a MiniSeq platform (Illumina, Inc.; San Diego, CA, USA). In brief, 100 µL cell culture supernatant from infected Vero E6 cells was combined with 300 µL TRIzol reagent (ThermoFisher Scientific) and RNA was purified using a Direct-zol RNA Miniprep kit (Zymo Research; Irvine, CA, USA). Purified RNA was further concentrated using an RNA Clean-and-Concentrator kit (Zymo Research), followed by quantification on a DeNovix DS-11 FX Fluorometer. RNA was converted to double-stranded cDNA, ligated then isothermally amplified using a QIAseq FX single cell RNA library kit (Qiagen; Hilden, Germany). Fragmentation and dual-index library preparation was conducted with an Illumina DNA Prep, Tagmentation Library Preparation kit. Average library size was determined using a Bioanalyser (Agilent Technologies; San Diego, CA, USA) and quantified with a Qubit 3.0 Fluorometer (Invitrogen; Carlsbad, CA, USA). Denatured libraries were sequenced on an Illumina MiniSeq using a 300-cycle Mid-Output Reagent kit as per the manufacturer’s protocol. Paired-end Fastq reads were trimmed for quality and mapped to the published sequence for the SARS-CoV-2 reference isolate Wuhan-Hu-1 (RefSeq: NC_045512.2) using CLC Genomics Workbench version 21 from which consensus sequences were generated. Stocks were confirmed to be free from contamination by adventitious agents by analysis of reads that did not map to SARS-CoV-2 or cell-derived sequences.

EpiAirway and EpiAlveolar tissue models were purchased from MatTek Corporation (Ashland, MA, USA). Both models were grown at the air-liquid interface, with cell compositions corresponding to the human airway and alveoli, respectively (Figure 1).

### 2.2. Titration of Samples

Samples were titrated using a 50% Tissue Culture Infectious Dose (TCID_50_) assay. In brief, samples were serially 10-fold diluted in DMEM supplemented with 2% FBS, 2 mM GlutaMAX supplement, 100 U/mL penicillin, and 100 μg/mL streptomycin, starting at a 1:10 dilution. Six replicate dilution series per sample were dispensed into wells of a 96-well plate (50 µL per well) into which 2 × 10^4^ Vero E6 cells/well in 100 µL volume were added. Plates were incubated at 37 °C/5% CO_2_ for four days before being assessed for the presence of cytopathic effect. TCID_50_ titres were calculated using the six replicates for each sample and the Spearman-Kärber method [25].

### 2.3. Preliminary Infections in Tissue Models

Infections were performed in the EpiAirway and EpiAlveolar tissue models to establish their susceptibility to SARS-CoV-2 infection, and their suitability for use in drug screening studies. Given their propensity to produce mucus, the apical side of the EpiAirway cells were washed twice with PBS prior to use. Four conditions (mock, virus-only, remdesivir-only (5 µM), and virus + remdesivir) were run in quadruplicate. A total of 5 µM remdesivir or the equivalent volume of DMSO was added to basal media of the appropriate wells 1 h prior to infection. Basal media containing DMSO or remdesivir was changed on Day 2 for the 72 and 96 h plates to ensure cell health was maintained.

Virus-containing inoculum was prepared by diluting SARS-CoV-2 Delta stock 1:10 in model-specific medium. Mock inoculum was composed of medium without virus added. A total of 100 µL or 200 µL appropriate inoculum was added to the apical side of the EpiAirway and EpiAlveolar models, respectively, for an effective multiplicity of infection (MOI) of approximately 0.01. After addition of inoculum, the cells were incubated for 1 h at 37 °C/5% CO_2_ before 300 µL PBS was added to the apical side of each tissue model followed by removal of all inoculum and wash.

For sample harvest, basal medium and apical wash samples were collected at 24, 48, 72, and 96 h post-infection. Apical wash samples were generated by applying 500 µL PBS to the apical side of each tissue model, incubating for 30 min at 37 °C/5% CO_2_, followed by removal into 2 mL Sarstedt tubes (Sarstedt; Mawson Lakes, SA, Australia). Both basal medium and apical wash samples were stored at −80 °C until titration.

### 2.4. Drug Selection, Procurement, and Preparation

Prospective drugs were down-selected from the Compounds Australia Open Drug collection using a set of filters as described previously [15]. The filters selected for the drugs included previous information about SARS-CoV-2 antiviral activity, approval status, and safety. Using this down-selection, the top ten drugs were selected for preliminary screening (Table 1), with an additional two drug analogues selected for secondary screening. To allow the scientific community to drive their own drug repurposing studies, a user-friendly web interface (http://www.covirx.org/) has been established, providing data for approximately 8000 compounds [26].

The selected drugs, along with remdesivir, molnupiravir, and nirmatrelvir (PF-07321332) for use as controls, were obtained from Selleck Chemicals (Houston, TX, USA) or Sigma Aldrich (St Louis, MO, USA). Details of the drugs and controls chosen for the study such as original indication, catalogue number, solubility and purity are summarised in Table 1, with detailed description of each in Table 2 of the Results section (Section 3.2).

Where possible, drugs were obtained pre-dissolved as 10 mM stocks in DMSO. For drugs not available in this format, 10 mM DMSO stocks were prepared and sterilised by filtration through a 0.22 µm syringe filter. As ondansetron was insoluble in DMSO, it was dissolved in 10 mM HCl and then filter sterilised.

### 2.5. Drug Screening Using the EpiAirway Tissue Model

Preliminary drug screening was performed using the EpiAirway tissue model. To maximise the number of drug-concentration combinations that could be tested, each drug-concentration combination was tested in singlicate, with controls (mock, virus-only, positive toxicity (100 µM rotenone), and 10 mM HCl toxicity) run in biological triplicates. The 10 mM DMSO stocks of drugs were diluted to the target concentrations of 25, 10, 2, 0.4, 0.08, and 0.016 µM in 5 mL EpiAirway medium and then added to the basal side of the tissue model 1 h prior to infection. To exclude drug-induced cytotoxicity-related effects at higher concentrations, wells treated with 10 µM of each drug, but not infected, were included in the experimental design.

Virus-containing inoculum was prepared by diluting SARS-CoV-2 Delta stock 1:10 in model-specific medium. Mock inoculum was composed of medium without virus added. An amount of 100 µL appropriate inoculum was added to the apical side of the EpiAirway for an effective MOI of approximately 0.01. After addition of inoculum, the cells were incubated for 1 h at 37 °C/5% CO_2_ before 300 µL PBS was added to the apical side of each tissue model followed by removal of all inoculum and wash.

For sample harvest, basal medium and apical wash samples were collected at 48 h post-infection. Apical wash samples were generated by applying 500 µL PBS to the apical side of each tissue model, incubating for 30 min at 37 °C/5% CO_2_, followed by removal into 2 mL Sarstedt tubes (Sarstedt; Mawson Lakes, SA, Australia). Both basal medium and apical wash samples were stored at −80 °C until titration.

Secondary drug screening was performed in the airway model using the three drugs that showed most promise in the preliminary screening (fluvoxamine, everolimus, and pyrimethamine), along with two analogues of drugs used in the preliminary screen (aprepitant and sirolimus). Three control drugs (remdesivir, molnupiravir, and nirmatrelvir/PF-07321332) were also run. As with the preliminary drug screening, each drug-concentration combination was tested in singlicate, with controls (mock, Delta virus-only, Omicron virus-only) run as biological triplicates. The 10 mM DMSO stocks of drugs were diluted to the target concentrations of 10, 4, 1, 0.4, and 0.08 µM for remdesivir and nirmatrelvir (PF-07321332), and 25, 10, 2, 0.4, and 0.08 µM for the rest, in 5 mL EpiAirway medium, which was then added to the basal side of the tissue model 1 h prior to infection. Against Omicron, additional treatment conditions were included combining 1 µM remdesivir with 10, 4, 1, or 0.4 µM pyrimethamine to determine whether a combinatorial effect could be observed.

Virus-containing inoculum for Delta was prepared by diluting SARS-CoV-2 Delta stock 1:10 in model-specific medium, while Omicron inoculum was prepared by dilution of SARS-CoV-2 Omicron stock 1:20 in medium. Mock inoculum was composed of medium without virus added. An amount of 100 µL appropriate inoculum was added to the apical side of the EpiAirway for an effective MOI of approximately 0.01. After addition of inoculum, the cells were incubated for 1 h at 37 °C/5% CO_2_ before 300 µL PBS was added to the apical side of each tissue model followed by removal of all inoculum and wash.

Toxicity of the drugs at 10 µM concentration (as well as 10 µM HCl for the Ondansetron stock), relative to the 100 µM rotenone positive toxicity controls was performed using a CyQUANT LDH Cytotoxicity Assay (ThermoFisher Scientific; Scoresby, VIC, Australia) with 1:100 dilution of the apical wash and basal media samples, following the manufacturer’s protocol.

## 3. Results

### 3.1. Preliminary Infections of EpiAirway and EpiAlveolar Tissue Models

Preliminary infections with SARS-CoV-2 Delta were performed in quadruplicate with mock, virus-only, remdesivir-only, and virus + remdesivir conditions. Basal medium and apical wash samples were collected at 24, 48, 72, and 96 h post-infection for virus titration.

Apical wash titres are presented in Figure 2. None of the basal media samples had detectable virus. Peak virus titres for the virus-only wells of the EpiAirway and EpiAlveolar models occurred on Day 3 post-infection with average titres of 4.7 × 10^6^ TCID_50/mL_ and 2.9 × 10^5^ TCID_50/mL_, respectively. In general, there was greater variation amongst the replicate samples in the EpiAlveolar model than the EpiAirway model, however both models supported effective virus replication. Furthermore, 5 µM remdesivir effectively suppressed virus growth in both tissue models.

### 3.2. Rationale behind the Selection of Candidate Drugs

Remdesivir has been in use since 2020, after it received approval status from FDA to treat COVID-19 [53]. It is a nucleoside analogue that truncates viral genomes during replication and introduces mutations [10]. Amongst several anti-viral agents, remdesivir has been reported to have an IC_50_ value around 8 µM and was able to reduce the viral load by approximately 30-fold at 1.5 µM by other researchers using in the EpiAirway model [54]. In addition, despite being relatively less potent in clinical trials, molnupiravir, another nucleoside analogue, was used as the second control drug in the preliminary drug screening experiment [55]. For the secondary drug screening experiment, a third control drug, nirmatrelvir (PF-07321332) was included. This drug is an inhibitor of the SARS-CoV-2 main protease and has been licenced for use as a COVID-19 therapy in humans [31,56].

As discussed in the materials and methods (Section 2.4), the test compounds were selected based on the down-selection of drugs from a library of approximately 8000 compounds. This down-selection generated a short-list of 214 compounds. After applying sequential filters, ranking methodologies, and in-depth pharmacological analysis, the top ten drugs were selected for testing within the EpiAirway model [15]. Salient features of the selected drugs, such as their C_max_, protein binding, etc, are described in Table 2.

### 3.3. Primary Testing of Candidate Drugs in the EpiAirway Tissue Model

Testing of the top ten primary candidate drugs was performed using the EpiAirway model. This model was selected due to it being more resilient than the EpiAlveolar model (the EpiAirway model better-tolerated shipment from the US to Australia and could be cultured for longer post-receipt compared to the EpiAlveolar model), as well as generally having more reproducible titres between replicates. Basal medium and apical washes were collected at 48 h post-infection to allow for the greatest virus growth without having to change the basal media, which could have impacted the study due to the administration of a *de facto* second dose of drug. 

As observed with the preliminary infections, virus was not detectable in the basal medium in any of the wells. Titres for the apical washes are summarised in Figure 3. For the control drugs, remdesivir reduced virus titres by 54-fold at 2 µM and 5400-fold at 10 µM, while molnupiravir only reduced virus titres (to below the limit of detection (BLoD)) at a 25 µM concentration. L-cycloserine, probenecid, ondansetron, cyclizine, lapatinib, and cetirizine did not reduce virus titres at any of the concentrations tested (Figure 3). By contrast, reduction in virus titre was observed at the highest concentrations for fluvoxamine (35-fold at 10 µM; BLoD at 25 µM), pyrimethamine (80-fold at 25 µM), everolimus (170-fold at 10 µM; BLoD at 25 µM), and rolapitant (5370-fold at 10 and 25 µM). These drugs compared favourably with molnupiravir but required higher concentrations than remdesivir for an equivalent reduction in virus titre. It is worth noting that 25 µM rolapitant exhibited some potential toxicity, as evidenced by apparently floating cells/cell debris in both the basal media and apical wash samples from sample wells.

Toxicity assays performed on apical wash samples from the 10 μM drug-only wells showed no significant toxicity at this concentration for any of the drugs (Figure 4). 

### 3.4. Secondary Testing of Candidate Drugs in the EpiAirway Tissue Model

To confirm results from the primary drug testing experiment, and to test two analogues of drugs that appeared to show some efficacy in said experiment (rolapitant and everolimus, analogues of rolapitant and sirolimus, respectively), fluvoxamine, everolimus, pyrimethamine, aprepitant, and sirolimus were tested using the EpiAirway model against both the Delta and Omicron (BA.1.1) VOC of SARS-CoV-2. Titrations were performed on the apical washes, and are presented in Figure 5 for both Delta and Omicron VOC. In this experiment, there appeared to be more variability between wells than observed with the primary screening experiment, particularly with Delta, as observed by the wider standard deviation lines for the virus-only controls. This was not unexpected as these wells were differentiated as a different batch to those used in the first experiment, and reflect the variability frequently observed with primary cells compared to cell lines. Against Delta VOC (Figure 5A), all three control drugs showed strong antiviral efficacy, with complete inhibition of virus replication at 1–4 µM. Most of the test drugs showed 10–100-fold reduction in virus titre at 25 µM, except for pyrimethamine, however the strength of the reduction was somewhat masked by inter-well variability. Against Omicron VOC (Figure 5B), there was strong activity with the control drugs, and there appeared to be a more pronounced antiviral effect of fluvoxamine and aprepitant at the higher concentrations (BLoD with 25 µM fluvoxamine; 75-fold reduction with 10 µM aprepitant), although minimal reduction in virus load was observed with the other test drugs. No toxicity was observed with the drug concentrations used in this experiment.

## 4. Discussion

Building upon in silico downselection of licenced drugs described in MacRaild et al. (2022), this study demonstrates the utility of human ex vivo/3D tissue models for the screening of drugs against emerging infectious diseases [15]. Considering the high cost and global distribution inequities of licenced antivirals for COVID-19 (such as remdesivir, molnupiravir, and nirmatrelvir), it is important to explore the antiviral potential of drugs licenced for other conditions. By broadening our antiviral arsenal, equitable access to effective treatments can be achieved globally, and not just in high-income countries [8,11,28,30,31,57,58].

As expected, remdesivir, molnupiravir, and nirmatrelvir showed strong inhibition of both Delta and Omicron (BA.1.1) VOC in the EpiAirway model. By contrast, only a handful of the test drugs showed anti-SARS-CoV-2 activity at the concentrations tested. Treatment of COVID-19 patients with fluvoxamine, a selective serotonin reuptake inhibitor, has shown clinical efficacy against severe infection, although true antiviral activity (as opposed to immune modulatory activity) has not been confirmed [59,60]. The results presented in this study suggest that fluvoxamine affect virus replication at concentrations >10 µM, although the antiviral activity of fluvoxamine is poorly characterised. Potential explanations include its lysosomotropic property and interference with viral entry by inhibiting the activity of acid sphingomyelase to convert sphingomyelin to ceramide and sphingosine [61]. The effective antiviral concentration observed in this study (>10 µM) means that direct antiviral activity in patients may be limited as such concentrations are unlikely to be encountered in clinical use, where plasma concentrations of the drug are usually around 0.3 µM following treatment [62]. It appears that the more likely explanation for clinical efficacy of fluvoxamine in COVID-19 lies in its anti-inflammatory activity through the decrease in endoplasmic reticulum stress responses and cytokine production via antagonistic effects on the sigma-1-receptor [63].

Aprepitant, a neurokinin-1 receptor antagonist used as an antiemetic in patients undergoing chemotherapy, showed efficacy against both Delta and Omicron VOC with >10-fold reductions in virus titre at concentrations ≥10 µM. In silico molecular docking analysis of SARS-CoV-2 proteases suggested that rolapitant (an analogue of aprepitant tested in the primary screen) and ondansetron could serve as inhibitors of the main protease (Mpro; targeted by nirmatrelvir) [64]. The relatively high concentration at which aprepitant showed antiviral activity means that direct clinical antiviral efficacy is unlikely as the maximum plasma concentration for aprepitant is approximately 2.8 µM–hence an effective dose is unlikely to be achieved within the patient [32].

The effect of pyrimethamine and everolimus on SARS-CoV-2 growth in the EpiAirway model was variable. Both drugs showed >100-fold reductions in virus titre at 25 µM in the primary drug screening experiment, but this activity was not repeated in the secondary drug experiment. The reason for this variation in efficacy is unclear. Pyrimethamine is an inhibitor of dihydrofolate receptor (DHFR), involved in the biosynthesis of nucleotides used in viral genome replication [65]. Accordingly, the impact of this drug may depend on the availability of free nucleotides within the infected cell. Everolimus, as well as its analogue, sirolimus, target the mTOR pathway which is known to be dysregulated during SARS-CoV-2 infection [66]. The use of sirolimus for the treatment of COVID-19 has entered Phase I clinical trials, but as yet no results have been published [67].

A few of the drugs tested showed a lack of efficacy in this study that contradicts previous studies. Lapatinib, an inhibitor of the ErbB family of receptor tyrosine kinases, was shown to inhibit SARS-CoV-2 growth in human lung organoids with an EC_50_ of 0.4 µM; however in this study, no inhibition was observed even at the highest drug concentration (25 µM) [41]. Probenecid, a uricosuric agent used in the treatment of gout, has contradictory anti-SARS-CoV-2 activity data in the literature. Murray et al. (2021) indicated that probenecid could inhibit SARS-CoV-2 replication in Vero cells and non-differentiated normal human bronchoepithelial cells, as well as a reduction in virus load in infected hamsters; however, Box et al. (2022) demonstrated a lack of antiviral activity in both Vero cells and hamsters [47,68]. Our results agree with the latter study, with no observable reduction in virus growth in the presence of the drug.

The screening of approved drugs for repurposing against new and emerging diseases is no mean feat. With nearly 8000 compounds licenced for use in humans, a harmonisation of approaches and coordination of efforts is appropriate to minimise duplication, reduce false positives/negatives, and make best use of limited resources. The website https://www.covirx.org/ was developed to allow for the sharing of such information [26]. This study has focussed on the evaluation of a small number of approved drugs using a commercially available ex vivo/3D tissue model system with a view to bridging the gap between in vitro and in vivo models. Most screening studies use cell lines, as they are easy to handle and can be used for high-throughput analysis; however, they frequently overstate efficacy as all the cells are in direct contact with the drugs by virtue of being bathed in media. Ex vivo/3D tissue models, particularly those grown at the air-liquid interface, are more representative of conditions within the body where, in some tissues, local concentrations of drugs may be lower due to strong intercellular attachments resulting in poor drug permeation. Moreover, as these models are derived from primary human cells, they reflect more accurately the variability in responses observed between individuals.

It is also important to note that drugs may show efficacy in one tissue model and not in another depending on the importance of the drug target for virus replication in that particular model. As such, a range of tissue models (e.g., cardiac, intestinal, lung, etc, for COVID-19) should be trialled before firm conclusions are drawn. A further critical consideration for drug repurposing is the determination of whether the effective concentration for antiviral activity lies below the maximum plasma concentration (C_max_) for the drug in humans. As mentioned above, direct antiviral activity of fluvoxamine and aprepitant in the EpiAirway model occur at concentrations significantly higher than can be achieved in patients following administration of typical therapeutic doses. Accordingly, their clinical efficacy (at least from an antiviral perspective) should be further investigated using tissue models of other important sites of infection, such as cardiomyocytes and intestine. In the present study, an attempt was made to investigate the efficacy of two drugs in combination (remdesivir and pyrimethamine). Such studies are recommended as the potency of combination drugs (particularly those with different mechanisms of action) can be synergistic, and may reduce the risk of resistance [69].

A final consideration when presenting in vitro efficacy data for readily available drugs is the potential for misuse. In the well-publicised cases of ivermectin and hydroxychloroquine, preliminary in vitro efficacy data (subsequently retracted) was exploited by some individuals to promote alternative agendas even after clinical studies have demonstrated a lack of efficacy in patients and, in some cases, that the use of these drugs can cause more harm than good [70,71,72,73,74]. Although this study demonstrated the antiviral activity of several licenced drugs against the Delta and Omicron VOC in 3D tissue models, results should not be used to support their prescription outside the context of a well-established clinical trial [75].

## Figures and Tables

**Figure 1 viruses-14-02417-f001:**
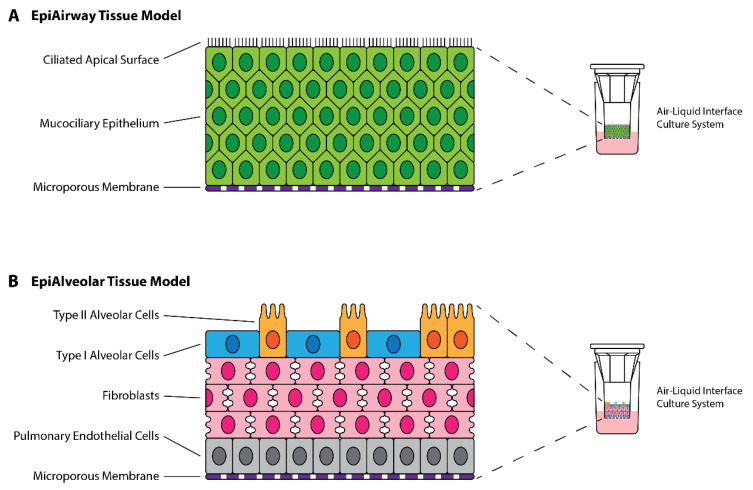
Structure of MatTek EpiAirway and EpiAlveolar 3D Tissue Models. (**A**) EpiAirway tissue model is composed of mucocilliary epithelial cells derived from normal human tracheal/bronchial epithelial cells (NHBE) differentiated at the air-liquid interface. (**B**) EpiAlveolar tissue model is composed of alveolar cells, fibroblasts and pulmonary endothelial cells derived from normal human alveolar epithelial cells (NHAE), normal human pulmonary fibroblasts (NHPF), and normal human pulmonary endothelial cells (NHPE) differentiated at the air-liquid interface. Colours are associated with cell types as labelled.

**Figure 2 viruses-14-02417-f002:**
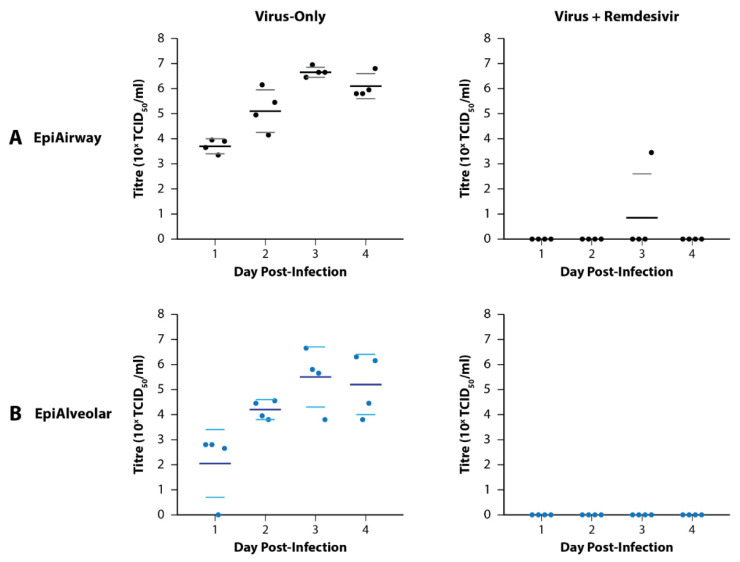
Growth of SARS-CoV-2 Delta in MatTek EpiAirway and EpiAlveolar 3D Tissue Models. (**A**) Growth in the EpiAirway tissue model peaked on Day 3 post-infection with relative reproducibility. An amount of 5 mM remdesivir was effective at preventing detectable infection in all but one of the replicates on Day 3. (**B**) Growth in the EpiAlveolar tissue model also peaked on Day 3 post-infection, but with more variability between the replicates than EpiAirway. A total of 5 µM remdesivir was effective at preventing detectable infection in all the samples. Dots represent titres from biological quadruplicate samples coloured by tissue type. For horizontal lines, the large dark lines represent the mean titre of the replicates, while the smaller, lighter lines represent the standard error of the mean (SEM).

**Figure 3 viruses-14-02417-f003:**
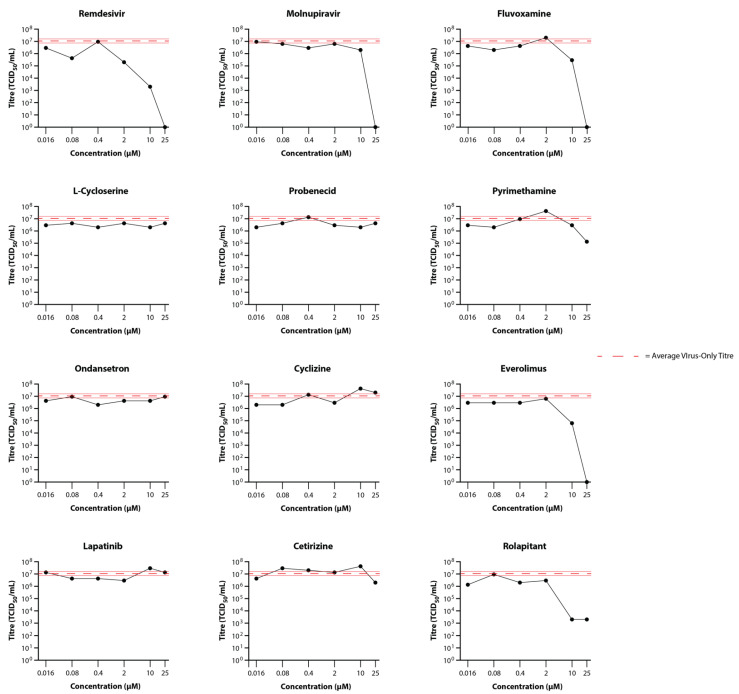
Primary Screening of Selected Drugs on the Growth of SARS-CoV-2 Delta in MatTek EpiAirway Tissue Model. Titres in the EpiAirway tissue model in the presence of 25, 10, 2, 0.4, 0.08, or 0.016 µM control (Remdesivir and Molnupiravir) or test (Fluvoxamine, L-Cycloserine, Probenecid, Pyrimethamine, Ondansetron, Cyclizine, Everolimus, Lapatinib, Cetrizine, and Rolapitant) drugs were determined by titrating the apical wash samples collected after 48 hr (black dots). The Red dashed line represents the average titre (±SD) from the biological triplicate Delta virus-only control samples.

**Figure 4 viruses-14-02417-f004:**
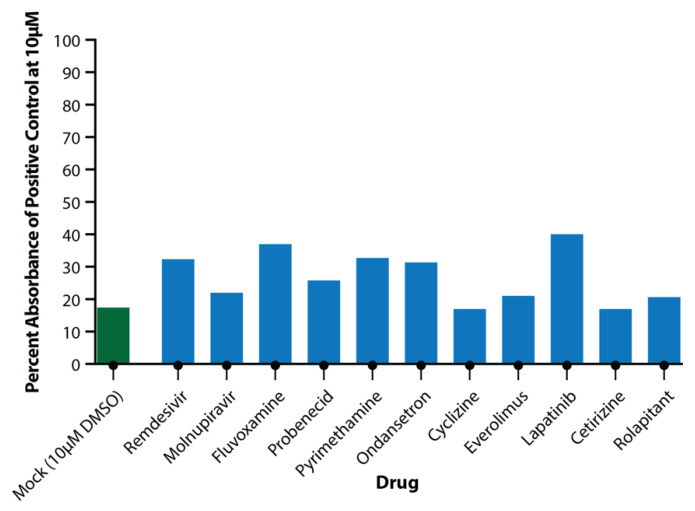
Toxicity of Drugs at 10 μM Concentration. Apical wash samples were collected and analysed in triplicate for LDH release from cells. Absorbance was compared to positive toxicity control (100 μM rotenone) and plotted as percentage of positive toxicity control for DMSO (green) and test drugs (blue).

**Figure 5 viruses-14-02417-f005:**
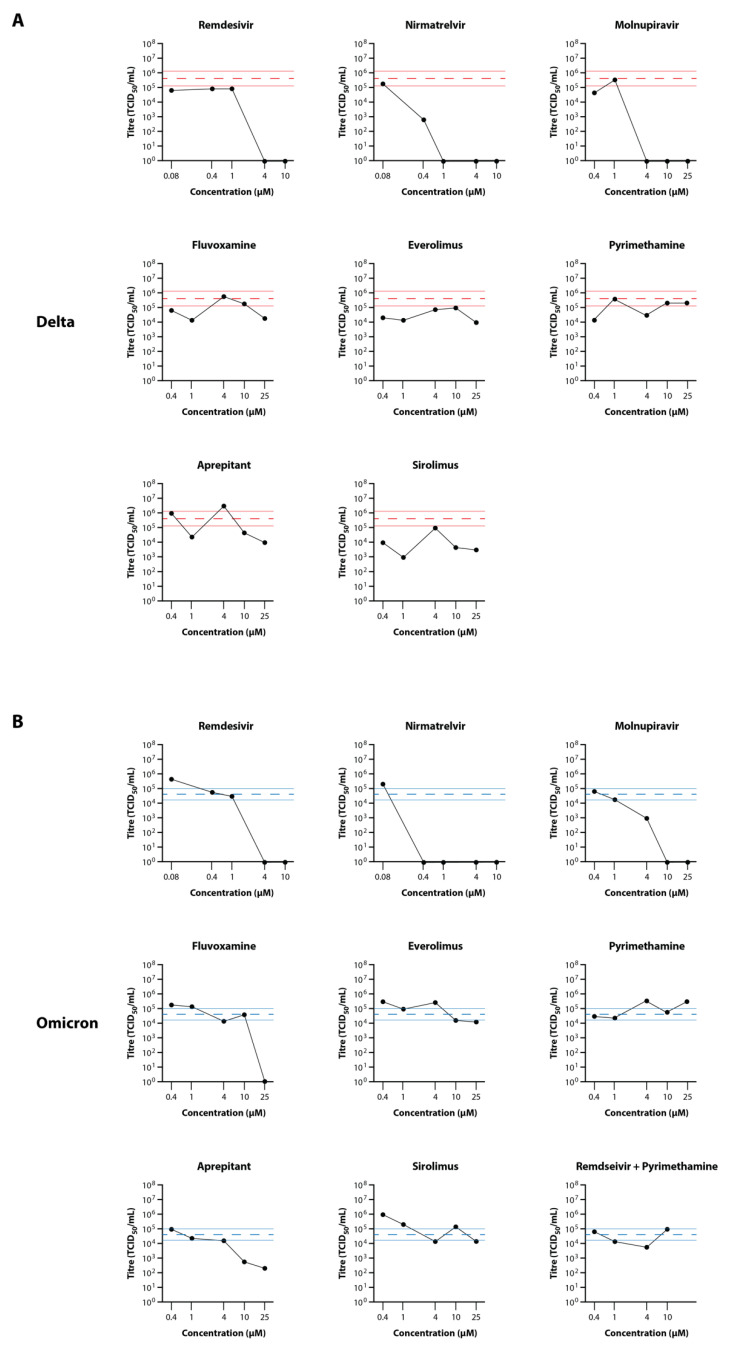
Secondary Screening of Selected Drugs on the Growth of SARS-CoV-2 Delta and Omicron in MatTek EpiAirway Tissue Model. Titres of SARS-CoV-2 Delta (**A**) and Omicron (**B**) in the EpiAirway tissue model in the presence of 10, 2, 0.4, or 0.08 µM remdesivir and nirmatrelvir or 25, 10, 4, 1, or 0.4 µM molnupiravir, fluvoxamine, everolimus, pyrimethamine, aprepitant, or sirolimus were determined by titrating the apical wash samples collected after 48 hr (black dots). The Red dashed line represents the average titre (±SD) from the biological triplicate Delta virus-only control samples, while the Blue dashed line represents the average titre (±SD) from the biological triplicate Omicron virus-only control samples.

**Table 1 viruses-14-02417-t001:** Test Drugs and Control Drugs Used in this Study.

Drugs	Original Indication	Catalogue Number	Purity	Solubility
Remdesivir	Antiviral	S8932	99.38%	DMSO, Ethanol
Molnupiravir	Antiviral	S0833	99.89%	DMSO, Water, Ethanol
Nirmatrelvir (PF-07321332)	Antiviral	S9866	99.82%	DMSO, Water, Ethanol
Aprepitant	Antiemetic	SML2215	>98%	DMSO, Ethanol
Cyclizine	Antiemetic	S0897	98.17%	DMSO, Water, Ethanol
Cetirizine	Antihistamine	S1291	99.54%	DMSO, Water
Everolimus	Immunosuppressant, Anti-Cancer	S1120	99.69%	DMSO, Ethanol
Fluvoxamine	Antidepressant	S1336	99.80%	DMSO, Ethanol
Lapatinib	Anti-Cancer	S2111	99.86%	DMSO
L-Cycloserine	Antibiotic	S3945	--	DMSO, Water
Ondansetron	Antiemetic	S1996	99.35%	0.5 M NaOH, 10 mM HCl
Probenecid	Anti-Gout	S4022	99.63%	DMSO, Ethanol
Pyrimethamine	Antiparasitic	S2006	100%	DMSO
Rolapitant	Antiemetic	S5476	99.67%	Ethanol
Sirolimus	Immunosuppressant	37,094	--	DMSO, Ethanol

DMSO = Dimethyl Sulphoxide; NaOH = Sodium Hydroxide; HCl = Hydrochloric Acid.

**Table 2 viruses-14-02417-t002:** Salient Features of Selected Drug Candidates.

Drug	Class	Description	Activity Data (against SARS-CoV-2)	C_max_ (µM)	Protein Binding (%)	Reference
Remdesivir	Antiviral	Nucleoside analogue used in the treatment of COVID-19	EC_50_ = 0.01 μM in human airway epithelial cells	--	83–93.6	[27]
Molnupiravir	Antiviral	Nucleoside analogue used in the treatment of COVID-19	IC_50_ = 0.08 μM in Calu-3 cells	--	--	[28,29]
Nirmatrelvir (PF-07321332)	Antiviral	Inhibitor of SARS-CoV-2 main protease (Mpro) used in the treatment of COVID-19	EC_50_ = 0.074 μM in Vero E6 cells	4.42	69	[30,31]
Aprepitant	Antiemetic	Neurokinin-1 receptor antagonist used in patients undergoing cancer chemotherapy	No activity data reported	2.80	>95	[32]
Cyclizine	Antiemetic	Histamine H1 antagonist used to treat motion sickness	EC_50_ = 10 μM against SARS-CoV-1 pseudovirus entry in Vero E6 cells	0.26	60–76	[33,34]
Cetirizine	Antihistamine	Second-generation antihistamine used to treat allergic reactions such as rhinitis, urticaria, dermatitis, etc	No activity data reported	0.80	93–96	[35]
Everolimus	Immunosuppressant	mTOR inhibitor used as an immunosuppressant to prevent rejection of organ transplants	Reduced SARS-CoV-2 gene and protein expression in Vero cells at 1 μM	0.19	74	[36,37]
Fluvoxamine	Antidepressant	Selective serotonin re-uptake inhibitor (SSRI) used for the treatment of obsessive compulsive disorder (OCD)	IC_50_ = 10.54 μM in HEK293T-ACE2-TMPRSS2 cells	0.28	77–80	[38,39]
Lapatinib	Anti-Cancer	Receptor tyrosine kinase inhibitor used for the treatment of breast cancer	EC_50_ = 0.7 μM in Calu-3 cells	4.18	>99	[40,41]
L-Cycloserine	Antibiotic	GABA transaminase inhibitor used to treat tuberculosis	Inhibition of replication in Vero E6 cells	830	None	[42,43]
Ondansetron	Antiemetic	5HT3 receptor antagonist used as an antiemetic	EC_50_ = 2.47 μM in Vero E6 cells; no protection of hamsters	0.43–0.66	73	[44,45]
Probenecid	Anti-Gout	Uricosuric agent used to decrease uric acid in the body	IC_50_ = 0.0013 μM in Normal Human Bronchoepithelial cells	521	75–95	[46,47]
Pyrimethamine	Antiparasitic	DHFR inhibitor used as an antiparasitic for the treatment of cystoisosporiasis and toxoplasmosis	58% inhibition of cytotoxicity in Caco-2 cells when administered at 10 μM	0.94	87	[48,49]
Rolapitant	Antiemetic	NK1 anatagonist used as an anti-emetic in patients undergoing chemotherapy	20% Inhibition of cytotoxicity in Vero E6	1.90	99.8	[50,51]
Sirolimus	Immunosuppressant	mTOR inhibitor used as an immunosuppressant to prevent rejection of organ transplants	Reduced SARS-CoV-2 gene and protein expression in Vero cells and airway cultures at 1μM	0.016–0.098	92	[37,52]

C_max_ = maximum plasma concentration; EC_50_ = 50% effective dose; IC_50_ = 50% inhibitory concentration; mTOR = mammalian target of rapamycin; GABA = Gamma Amino Butyric Acid; 5HT3 = 5 hydroxytryptamine; DHFR = Dihydrofolate Reductase; NK1 = Neurokinin 1.

## Data Availability

Any underlying data not presented can be provided by the corresponding author upon reasonable request.

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
