# Peer review of "Use of Human Lung Tissue Models for Screening of Drugs against SARS-CoV-2 Infection"

_viruses, 2022, doi:10.3390/v14112417_

Round 1

Reviewer 1 Report

This manuscript by McAuley AJ et al aims to demonstrate the usefulness of commercially available tissue models of human epithelial cells obtained from trachea/bronchi or from pulmonary alveoli, in order to screen drugs with potential anti-SARS-CoV-2 activity. The manuscript is well written, methods and results are clearly presented and the content is relevant for the research of SARS-CoV-2 antivirals.

I have some comments in order to improve the scientific content and comprehension of this article:

1.     In the ex vivo model presented in the manuscript, the virus is added to the apical side of the tissue (resembling the infection through the air) and all the drugs are added to the basal medium of the tissue model, resembling the effect that drugs would have on the tissue when administrated intravenously. However, there are some drugs that could be administrated orally or intranasally with a spray, and in this case the drugs would interact with the apical part of the model. For these drugs, it would be interesting and relevant to see their effect on SARS-CoV-2 infection when added to the apical side.

2.     The so called “preliminary” and “secondary” drug screening using the EpiAirway tissue model follow the same protocol, and only differ the virus tested (Delta in preliminary, Delta and omicron in the secondary) and some of the drugs tested. In the Materials&Methods section would be easier to follow if explaining the protocol only once, unifying the description and remarking the differences between these two tests/screenings.

3.     In the Results section 3.3, why the authors collect the basal and apical samples at 48h if the peak virus titer is reached at day 3 post-infection, as shown in Figure 2?

4.     The toxicity of tested drugs must be included in the manuscript. The toxic concentrations should be marked or excluded in the figures that show the results.

5.     There is an incoherence in the text (section 3.3): line 359 says that 25uM rolapitant exhibited some toxicity, but in line 370 says that any of the drugs showed significant toxicity and 10mM.

6.     In section 3.4, it would help the comprehension if authors explain that aprepitant and sirolimus are analogs of rolapitant and everolimus.

7.     Results in Figure 3 show Remdesivir and Molnupiravir are effective at 25uM and in Figure 4 they are effective at 4uM. Furthermore, the Fluvoxamine in Figure 3 is effective at 25uM and in Figure 4 it seems to not have effect on SARS-CoV-2 infection. What is the explanation for such variability? Given this variability between experiments, I recommend to perform duplicates for each condition/drug to be tested, in order to have more robust results.   

Specific comments:

Line 117-118, between SARS-CoV-2 and 3D models should be a dot to separate the sentences.

Some references lack information or have some mistake: 16, 28, 29, 32, 34, 37, 38 43, 50, 53, 59. Please revise and complete them.

Reviewer 2 Report

The authors aim to describe the use of commercially available human lung tissue models for screening of repurposed drugs against SARS-CoV-2 infection. However only one type of model have been used (tracheal/bronchial epithelium) to evaluate antiviral effect. It would have been interesting to compare pathophysiology and impact of Delta and Omicron variants on EpiAirway and ApiAlveolar models and potentially differential antiviral activities on these models. Furthermore, the models being prepared from primary human cells, interindividual variability should be discussed.

Despite the interesting anti SARS-CoV-2 activity of fluvoxamine, aprepitant, everolimus and sirolimus against Delta and Omicron variants reported, not enough novelty of the study justify a publication in Viruses. This work might be published in lower impact factor journal.   
